# Identification of Corrosion on the Inner Walls of Water Pipes Using a VGG Model Incorporating Attentional Mechanisms

Qian Zhao [1,2], Lu Li [1,*] and Lihua Zhang [1]

1   School of Communication and Information Engineering, Xi'an University of Science and Technology, Xi'an 710054, China
2   Xi'an Key Laboratory of Network Convergence Communication, Xi'an 710054, China
*   Correspondence: 20207223093@stu.xust.edu.cn

**Abstract:** To accurately classify and identify the different corrosion patterns on the inner walls of water-supply pipes with different morphologies and complex and variable backgrounds, an improved VGG16 convolutional neural network classification model is proposed. Firstly, the S.E attention mechanism is added to the traditional VGG network model, which can be used to distinguish the importance of each channel of the feature map and re-weight the feature map through the globally calculated channel attention. Secondly, the joint-loss-function method is used to improve the loss function and further improve the classification performance of the model. The experimental results show that the proposed model can effectively identify different pipe-corrosion patterns with an accuracy of 95.266%, higher than the unimproved VGG and AlexNet models.

**Keywords:** VGG16; SE attention mechanism; joint-loss function; corrosive classification recognition

## 1. Introduction

Water is the material basis for human life and the survival of all things, and is also an indispensable material resource for social production [1]. With the development of the economy, changes in consumption patterns, and improvements in urbanization level, people's demand for water resources is increasing rapidly, which is expected to accelerate considerably in the next 20 years. However, the waste of water caused by water leakage is a very serious problem [2].

According to the American Water Works Association (AWWA), the vast majority of water-supply pipes in the United States are gray cast-iron pipes and ductile iron pipes. As shown in Figure 1, more than 90% of the existing water-supply pipelines in China use metal pipelines. In recent years, the proportion of metal pipelines among the water supply pipelines under construction has been as high as 85%. These results show that in high-humidity environments, metal pipeline walls are prone to a series of electrochemical reactions with water environments, such as oxygen, water temperature, PH value, etc. Corrosion products, such as rust tumors, are formed on the pipeline walls. Although some anti-corrosion measures are taken in the water-supply network, corrosion and scaling still occur as the service life increases, as shown in Figure 2. The light-corrosion problem causes water-quality deterioration, affecting the health of users [3]. Heavy corrosion causes damage to the inner walls of metal pipelines, reduces the service life of pipelines, and causes pipeline leakage. According to the standard, the service life of water supply pipelines in our country is generally 30–50 years. Recently, the water-supply pipelines of many cities have entered the aging stage. Pipeline corrosion, aging, leakage, and other problems are unavoidable. Therefore, it is of great significance to study the types of pipeline corrosion and formulate effective maintenance strategies for ensuring the safe operation of water-supply networks.

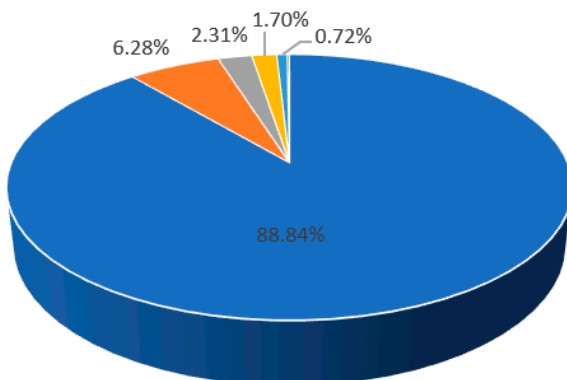

**Figure 1.** Proportion diagram for different pipe materials.

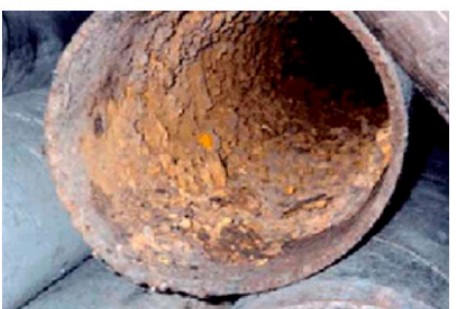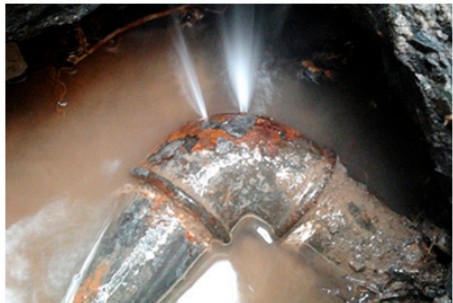

**Figure 2.** Pipeline corrosion and leakage diagram.

Existing pipeline inspection methods include leaky magnetic detection [4], ultrasonic inspection [5], and the direct assessment of external corrosion. However, these methods have limitations, including high equipment costs, limited inspection coverage, and the inability to detect small areas of pitting corrosion. There is, therefore, an urgent need for an efficient, low-cost inspection method for pipeline systems [6].

Among machine-learning methods, Medeiros et al. [7] proposed a model for classifying corroded and non-corroded surfaces using texture descriptors obtained from gray-scale co-occurrence matrices and image-color features. Hoang et al. [8] proposed an automated method for corrosion detection on the inner walls of water-supply pipes by combining an image texture feature extraction algorithm and a support vector machine classifier with differential pollination optimization. The method is based on a combination of an image-texture-feature-extraction algorithm and a support vector machine classifier with differential pollination optimization. This process can be used to classify and identify defects by designing a classifier model to identify corrosion on the pipe's inner wall. However, it features some problems. For example, features need to be created and extracted manually, but human definitions of features can be influenced by experience. Manually designed defect features are not resistant to variations in diversity, and recognition rates are significantly reduced when manually created algorithms cannot express the image's high-level features.

With the development of computer technology, deep-learning techniques have been widely used in the field of structural health monitoring and can be effectively used to study defects on internal and external surfaces of pipes or other metal surfaces [9], such as corrosion and cracks. Atha and Jahanshahi [10] used ZFNet and VGG16 for corrosion detection and the assessment of metal surfaces. Papamarkoua [11] used ResNets for corrosion detection on the surfaces of nuclear fuel tanks. Neither method considered the influence of feature importance on the detection results, and the detection accuracy could be improved. Kumar [12] achieved the automatic recognition of drainage-pipe cracking, cracks, corrosion, and other defect types through an improved AlexNet network, in which the recognition accuracy of corrosion was not high. Hassa et al. [13] proposed a

convolutional neural network (CNN)-based pipe-defect classification system. The above method only detects the presence or absence of corrosion on the pipeline without classifying or identifying different corrosion patterns, which lacks practicality for realistic water-supply pipelines in which different corrosion types exist. The accuracy rate needs to be improved.

Therefore, to achieve the accurate classification and recognition of different corrosion patterns of pipes, a VGG16 classification model incorporating an attention mechanism is proposed. The loss function is improved using a joint-loss function. Finally, the model is applied to the damage dataset of the pipe's inner wall to classify and recognize corrosion patterns of the inner wall in the water pipe.

## 2. Image Acquisition and Sample Set Production

### 2.1. Pipeline-Damage-Image Acquisition

This paper reports the use of an industrial endoscope acquisition platform to achieve the real-time acquisition of the corrosion images of the inner wall of the pipeline [14]. The acquisition platform is shown in Figure 3, and the acquired samples are shown in Figure 4.

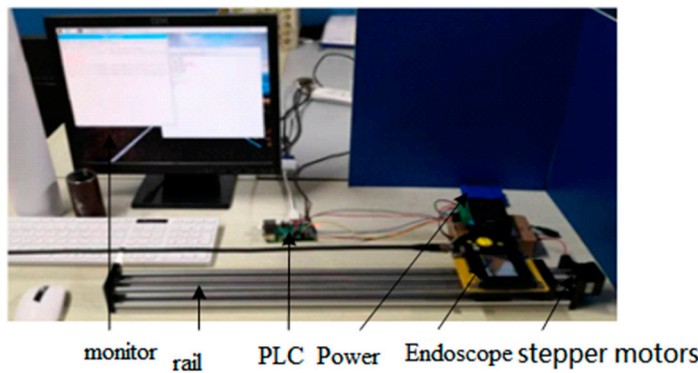

**Figure 3.** Endoscopic acquisition platform.

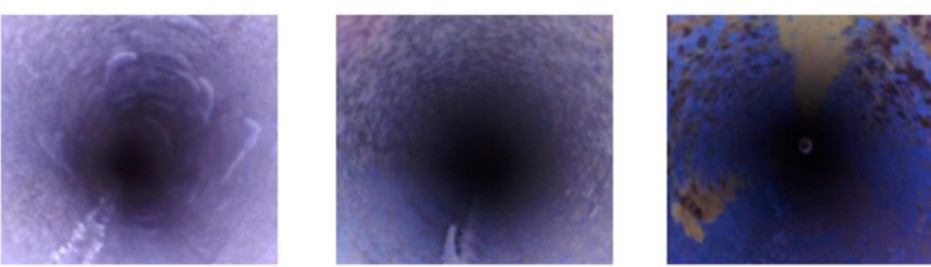

**Figure 4.** Partial image acquisition result map.

Due to the particular characteristics of the pipeline shape and acquisition platform, the acquired images are presented in a conical manner and contain three-dimensional information. At the same time, as the pipeline is often located underground or in a closed environment, the acquired images have a highly uneven distribution of grayscale information, and the outline and edges of the corrosion area are not unmistakable. They are not sufficiently prominent in the background. The difference between the grayscale values of the corrosion area and the background is small, which makes the subsequent feature extraction more difficult and affects the damage-image-feature-recognition results, reducing the damage recognition rate. Therefore, the need for image pre-processing, the pre-processing process is shown in Figure 5. The method of the pre-processing algorithm is as follows: firstly, using the cone-based bi-directional projection model of the original pipe image panoramic expansion to obtain two-dimensional image information [15], and secondly, using the bi-directional illumination estimation model modified Retinex algorithm to expand the image enhancement process to improve the image contrast [16].

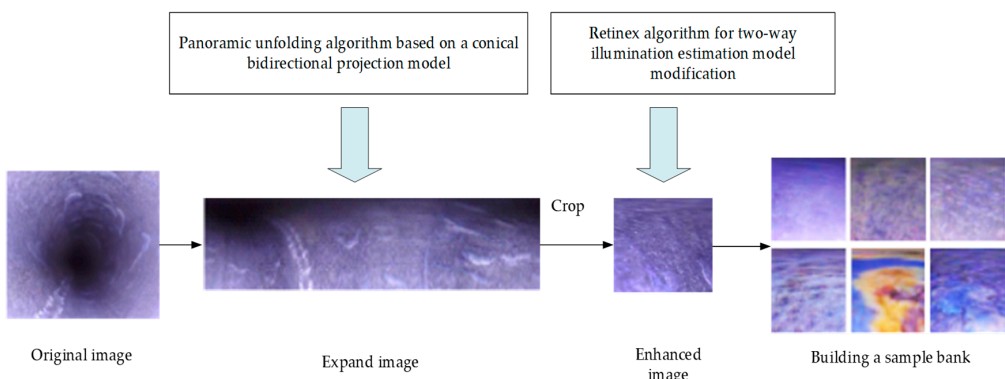

**Figure 5.** Image pre-processing process diagram.

## 2.2. Preparation of Experimental Sample Set

In this study, an industrial endoscope acquisition platform was used to obtain corrosion images of the pipe's inner wall, containing four corrosion patterns, as shown in Figure 6. Due to the small number of experimental samples, data enhancement was used to expand the samples [17]. Commonly used data enhancements include image flip, image noise addition, and image-brightness adjustment, as shown in Figure 7. The number of enhanced samples is shown in Table 1. The final dataset had a total of 6799 images, including 5954 images in the training set and 845 in the test set.

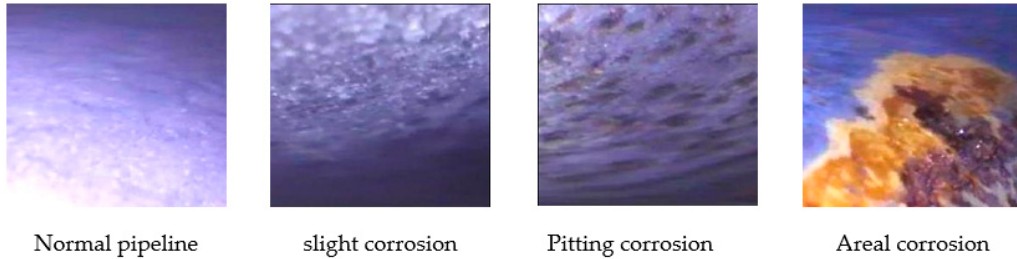

**Figure 6.** Sample-data presentation.

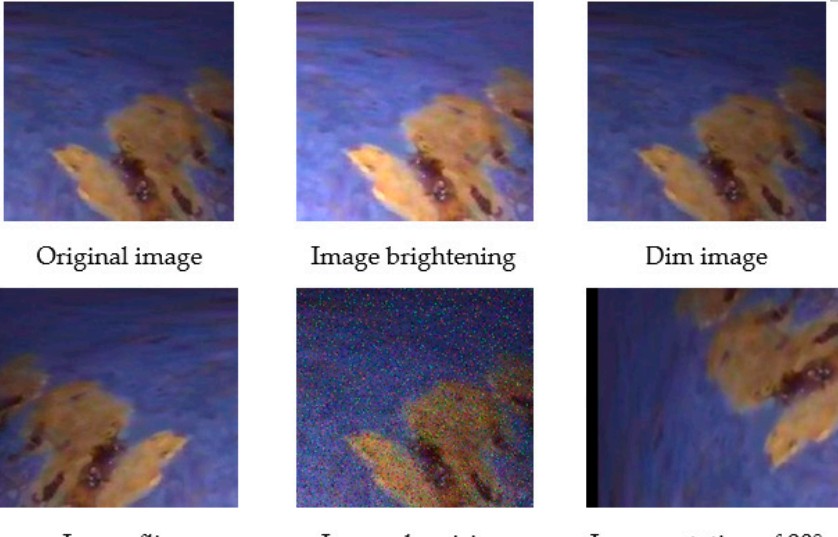

**Figure 7.** Data enhancement.

**Table 1.** Sample size before and after enhancement.

| Sample | Before Data Enhancement | After Data Enhancement |
| --- | --- | --- |
| Pitting corrosion | 783 | 1730 |
| Areal corrosion | 788 | 1739 |
| Slight corrosion | 790 | 1742 |
| Normal pipeline | 724 | 1588 |

## 3. Method

### 3.1. Basic Concept of Convolutional Neural Networks

In recent years, machine-learning-based algorithms have been widely used in structural damage detection [18]. Classification and recognition are very common problems in the field of machine learning. For this problem, there are different algorithms in machine learning, such as linear regression, logical regression, SVM, decision tree, neural network, etc. In this study, a convolutional neural network was used to recognize and classify the corrosion images.

The two most important ideas in convolution networks are sparse connection and parameter sharing. When the same convolution kernel is used on the input layer, the feature of parameter sharing can be obtained, which can greatly reduce the required parameters in the CNN model and reduce the computational complexity. CNNs generally consist of an input layer, convolution layer, pooling layer, full connection layer, and output layer.

The convolution layer mainly extracts the abstract features of the input data, the pooling layer reduces the dimensions of the images, and the main function of the full connection layer is classification.

### 3.2. Mathematical Principle of Convolution Neural Network

Convolutional neural networks are essentially a multi-level sensor that is non-linear. The input is mapped directly to the output. Its main features are weight sharing and local linking, which can greatly reduce the number of weights and the complexity of the model acquisition. The two core components of convolution neural networks are the convolution layer and the pooling network. They can make the loss-function transfer backward layer by layer and automatically adjust the weight value as little as possible through gradient descent according to the characteristics of the optimization and extraction of various input data. At the same time, the network continues to optimize and train repeatedly until the model of the convolution neural network finally converges. The network automatically adjusts the data-weight value of various output networks at any time. Finally, a convolutional neural network model is trained.

The algorithm process of convolution neural network implementation is as follows:

(1) Set $x^1$ as the initialization data input to the input layer for the first time, and $x^2$ as the output result of data $x^1$ after convolution operation in the first convolution neural network; $w^1$ is the parameter of the first layer of the neural network. At the same time, $x^2$ is input to the second layer of the network, and $x^3$ is output after convolution and pooling through the second layer of the network; $w^2$ is the parameter of the second layer of the neural network, which is calculated to the last layer of the network in this way, while $x^L$ can be set as its final output result and $w^{L-1}$ is the parameter of the last layer network. Finally, to determine differences between the output value and the predicted value and calculate the loss value of the network as $z$, the expression is:

$$z = L(x^L, y) \tag{1}$$

(2) The dimension of the output $x^L$ of the convolutional neural network is the same as the real value $y$, and the expression of the predicted value obtained after forward propagation is:

$$\mathrm{argmax}.x_i^L \tag{2}$$

(3)  Set the sample data inputted to the input layer for the $i$ th time by the convolutional neural network as $\left\{ \left( x^{(i)}, y^{(i)} \right) \right\}, 1 \leq i \leq N$. At the same time, set $W$ as the weight in each layer of the network, and $b$ as its offset in each layer. The output result is $f\left( x^{(i)} \middle| W, b \right)$. The loss function $J(W, b)$ can be calculated, and its mathematical expression is:

$$J(W, b) = \sum_{1}^{N} L\left( y^{(i)}, f\left( x^{(i)} \middle| W, b \right) \right) + \frac{1}{2} \lambda \|W\|_F^2 = \sum_{1}^{N} J\left( W, b; x^{(i)}, y^{(i)} \right) + \frac{1}{2} \lambda \|W\|_F^2 \quad (3)$$

$$\|W\|_F^2 = \sum_{l=1}^{L} \sum_{i=1}^{n^{l+1}} \sum_{l=1}^{n^l} W_{ij}^{(l)} \quad (4)$$

(4)  Carry out iterative training many times, and constantly update the parameter weights of the network, so as to minimize the loss function $J\left( W, b; x^{(i)}, y^{(i)} \right)$ of the network. Next, the calculation method of gradient direction descent is used to conduct systematic learning and design updating for all kinds of weight parameters and offset weight values of the whole network. Let the network weight of the $i$th iteration be $W^{(l)}$ and the network offset be $b^{(l)}$. The mathematical expressions of $W^{(l)}$ and $b^{(l)}$ at this time are calculated as follows:

$$W^{(l)} = W^{(l)} - \alpha \frac{\partial J(W, b)}{\partial W^{(l)}} = W^{(l)} - \alpha \sum_{i=1}^{N} \left( \frac{\partial J\left( W, b; x^{(i)}, y^{(i)} \right)}{\partial W^{(l)}} \right) - \lambda W \quad (5)$$

$$b^{(l)} = b^{(l)} - \alpha \frac{\partial J\left( W, b; x^{(i)}, y^{(i)} \right)}{\partial b^{(l)}} \quad (6)$$

(5)  In the process of forward nerve propagation, $z^{(l)}$ is used to directly represent the active activity state of neurons in layer $l$, and $a^{(l)}$ is used to represent the active state value of neurons in the first level after activation. Subsequently, the expression of neurons in layer $l$ is defined as:

$$z^{(l)} = W^{(l)} \cdot a^{(l-1)} + b^{(l)} \quad (7)$$

(6)  To calculate the partial derivative of the loss function $J\left( W, b; x^{(l)}, y^{(l)} \right)$ of the layer $l$ neural network to the neuron $z^{(l)}$ of the layer $l$, recorded as $\delta^{(l)}$, the mathematical expression is:

$$\delta^{(l)} = \frac{\partial J(W, b; x, y)}{\partial z^{(l)}} \quad (8)$$

The derivative of the transfer function of each layer can be obtained as follows:

$$\frac{\partial J(W, b; x, y)}{\partial W_{ij}^{(l)}} = tr \left[ \left( \frac{\partial J(W, b; x, y)}{\partial W_{ij}^{(l)}} \right)^T \frac{\partial z^{(l)}}{\partial W_{ij}^{(l)}} \right] \quad (9)$$

The partial derivative expression of neurons in layer $l$ state is:

$$\frac{\partial z^{(l)}}{\partial W_{ij}^{(l)}} = \frac{\partial \left( W^{(l)} \cdot a^{(l-1)} + b^{(l)} \right)}{\partial W_{ij}^{(l)}} \quad (10)$$

Next, calculate the partial derivative of the weight of the layer, the partial derivative of the layer offset is:

$$\frac{\partial J(W, b; x, y)}{\partial W_{ij}^{(l)}} = \delta_i^l \left( a^{(l)} \right)^T \quad (11)$$

$$\frac{\partial J(W, b; x, y)}{\partial b^{(l)}} = \delta^{(l)} \tag{12}$$

According to the above steps, it is possible to update the parameters of the network iteratively. Finally, a convolutional neural network is trained.

### 3.3. VGGNet

VGGNet won 2nd place in ImageNet image classification in 2014. VGG16 is one of the best VGG networks in classification performance, and its network structure is shown in Figure 8 [19].

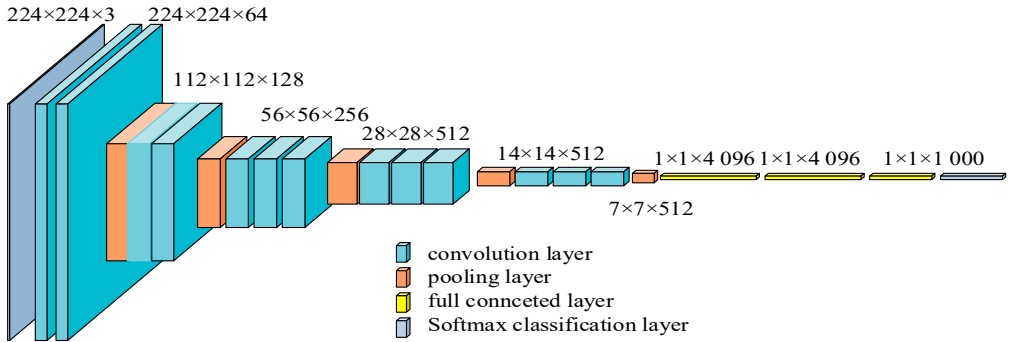

**Figure 8.** VGG16 network structure.

The VGG16 network structure consists of 13 convolutional layers and three fully connected layers for 16 weight layers. The input part of VGG16 is a 224 × 224 × 3 RGB image. After processing the entire convolutional network, the output structure is the probability that the input image belongs to each category. The model is divided into six parts overall.

The first five parts consist of multiple convolutional layers with a convolutional kernel size of 3 × 3, and the latter consists of 3 fully connected layers. Each convolutional layer in VGG16 has a convolutional step (stride) of 1 pixel, and the input and output dimensions are kept constant by a boundary padding (padding) of 1pixel. The pooling layer uses maximum pooling with a window size of 2 × 2. The number of channels in the three fully connected layers is 4096, 4096, and 1000 respectively, with 1000 representing the score of the input image in each of the 1000 classes. Since VGG-16 is trained on the Image Net dataset, the number of classifications is 1000. The last layer is the Softmax classification layer, which converts the scores into probabilities that the input image belongs to each class. To make the weights non-linear, ReLU is used in the convolution layer to conduct a non-linear transformation to accelerate the convergence of the network.

### 3.4. S.E. Attention Mechanism

In recent years, the attention mechanism has been widely used in deep learning tasks [20]. The attention mechanism can capture the differences in importance between each part of the input feature matrix and assign different weights to extract more critical and discriminative information, allowing the model to make more accurate judgments.

The essence of SE-Net is the use of a channel attention mechanism to automatically obtain the weights of each feature channel of an image using deep learning to enhance useful features and suppress useless features. The structure of SE-Net is shown in Figure 9. A feature matrix U with channel number c is obtained by applying a series of convolutional transformations to the input feature matrix X with channel number c'. The structure that implements the attention mechanism consists of Squeeze, Excitation, and Scale [21].

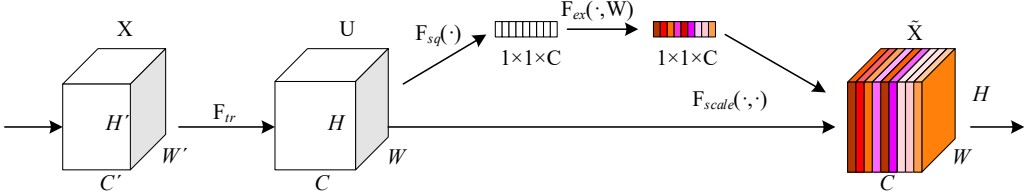

**Figure 9.** S.E. Module.

(1) Squeeze ($F_{sq}$) operation: This step pools the image's feature maps to obtain each channel's global features, as shown in Equation (13).

$$z_c = F_{sq}(u_c) = \frac{1}{H \times W} \sum_{i=1}^{H} \sum_{j=1}^{W} u_c(i,j) \tag{13}$$

(2) Excitation ($F_{ex}$) operation: This step is performed through two fully connected layers that generates the required weight information through the weights, which are obtained through learning and are used to model the relevance of the features needed for the display, as shown in Equation (14).

$$s = F_{ex}(z, W) = \sigma(g(z, W)) = \sigma(W_2 \delta(W_{1z})) \tag{14}$$

where $W_1$ is the first full-connection-layer operation, $W_2$ is the second full-connection-layer operation, $\delta$ is the activation function Relu, and $\sigma$ is the activation function Sigmoid.

(3) Reweigh ($F_{scale}$) operation: The weights obtained in the previous step are weighted to the original features by multiplying them channel by channel to complete rescaling of the original features in the channel dimension, as shown in Equation (15).

$$\widetilde{x}_c = F_{scale}(u_c, s_c) = s_c \cdot u_c \tag{15}$$

*3.5. Improved VGG16 Model*

3.5.1. SE-VGG16 Classification Model

The model makes more accurate judgments to extract more critical and discriminative information from the corrosion image. This paper proposes to add the attention mechanism module S.E. to the VGG16 network to improve it, resulting in an improved VGG16 network model suitable for the classification task of damage recognition in the inner wall of pipes.

A comparison of the structure of the model before and after the improvement is shown in Figure 10. The overall design of the improved model is as follows. The input image size is $224 \times 224 \times 3$; after the convolution layer of 64 convolution kernels is convolved twice and ReLU is activated, the output size is $224 \times 224 \times 64$. After the maximum pooling layer is pooled, the image size is halved, and the output size becomes $112 \times 112 \times 64$. After the convolution layer of 128 convolution kernels is convolved twice and ReLU is activated, the output size is $112 \times 112 \times 128$. After maximum pooling layer pooling, the output size becomes $56 \times 56 \times 128$. After 256 convolutional kernels of convolutional layer convolution three times, ReLU activation, the output size is $56 \times 56 \times 256$. After maximum pooling layer pooling, the output size becomes $28 \times 28 \times 256$. After 512 convolutional kernels of convolutional layer convolution three times, and ReLU activation, the output size is $28 \times 28 \times 512$. After maximum pooling layer pooling, the output size becomes $14 \times 14 \times 512$. After 512 convolutional cores of convolutional layer convolution three times, ReLU activation, the output size is $14 \times 14 \times 512$. After maximum pooling layer pooling, the output size becomes $7 \times 7 \times 512$. After S.E. network module, the output size is $7 \times 7 \times 512$. Next, flattening (Flatten) is performed to turn the data into a one-dimensional vector, $7 \times 7 \times 512 = 25088$. After a fully connected layer of $1 \times 1 \times 6$, the Dropout is 0.5 and, finally, the prediction result is output by the Softmax classifier.

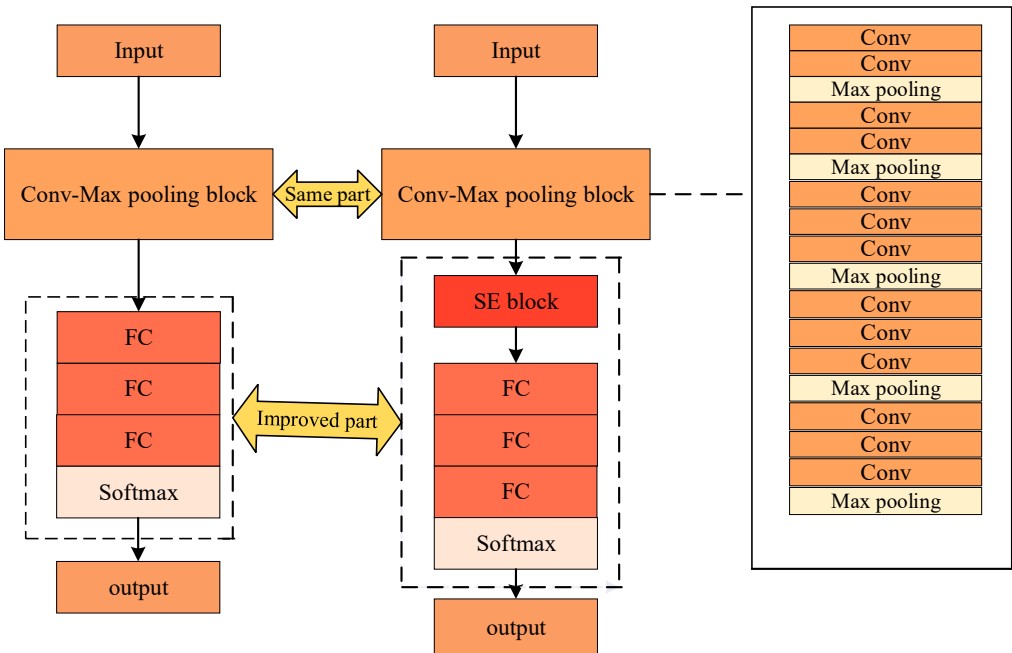

**Figure 10.** Before and after comparison of VGG16 model improvements.

### 3.5.2. Multi-Loss Function Fusion

The cross-entropy loss function is usually used for common image-classification problems to find the loss. Cross entropy represents the difference between the actual probability distribution and the predicted probability distribution [22]. The smaller the value of cross-entropy, the better the model prediction. The cross-entropy loss function is defined in Equation (16). However, in the problem of classifying pipeline-corrosion patterns, the situation of corrosion-free and lightly corroded pipelines is very similar, resulting in little feature differentiation.

$$C = -\sum_{x=1}^{N} \left[ y_{(x)} \log \widehat{y}_{(x)} + \left( 1 - y_{(x)} \right) \log \left( 1 - \widehat{y}_{(x)} \right) \right] \tag{16}$$

where $N$ is the number of samples in the batch, $x$ is the index symbol for the sum, $y$ is the label value, and $\widehat{y}$ denotes the actual output.

The advantage of the center loss function is that it can learn features with smaller intra-class distances, thus reducing intra-class variation and, to some extent, increasing inter-class variability, thereby improving classification accuracy [23]. The definition of the central loss function is given in Equation (17):

$$L_c = \frac{1}{2} \sum_{i=1}^{n} ||x_i - C_{yi}||_2^2 \tag{17}$$

where $n$ is the number of batch samples, $x_i$ denotes the $i$ feature of category $y_i$, and $C_{yi}$ denotes the centroid of category $i$ features.

Therefore, the cross-entropy and center loss function's joint loss function is used. Including center loss ensures that the feature distance within a class is minimized, allowing samples from the same type to be close to the feature center of the corresponding class, making training more accessible and facilitating optimization. The joint loss function is calculated as in Equation (18).

$$L = C + \lambda Lc = -\sum_{x=1}^{n} \left[ y \log \widehat{y}_x^l + (1 - y) \log \left( 1 - \widehat{y}_x^y \right) \right] + \frac{\lambda}{2} \sum_{i=1}^{n} ||x_i - C_{yi}||_2^2 \tag{18}$$

The fusion coefficient $\lambda$ ranges from 0 to 1, and the fusion coefficient that gives the highest classification accuracy was found to be 0.4 through several experiments.

### 3.6. Hyperparameter Optimization

The success of convolutional neural networks depends on the selected structure and super parameters. A hyper parameter is a parameter that needs to be set manually before model training and cannot be obtained through sample-data estimation and model training. For different network structures and data sets, the optimal super parameters are different, and the performance of the model needs to be improved by optimizing the super parameters. The generalization performance of CNN model often depends on the selection of its preset super parameters. Bayesian optimization algorithm is a global optimization Algorithm 1 [24]. Through the given black-box objective function, the sample points are continuously added, and the posterior distribution of the objective function is updated until the posterior distribution basically conforms to the real distribution. Only a few iterations are required to obtain the ideal solution. Therefore, this paper uses Bayesian optimization algorithm to optimize the super parameters of the network in this paper.

The process of Bayesian optimization algorithm for super parameter estimation is shown below. First, starting from the initial super-parameter selection, according to the existing sample points D, we can obtain a new super-parameter vector $x_{t+1}$, which is more likely to obtain the optimal value in order by optimizing the collection function $\alpha(x)$. Next, function evaluation is performed on the new samples to obtain new generalization performance $y_t = f(x_t) + \varepsilon_t$. The Bayesian optimization algorithm used in this paper selects the Gaussian process as the probabilistic proxy model and UCB as the collection function.

---

**Algorithm 1:** Labeling Bayesian optimization algorithm

---

Input: Agent model $f$, collection function $\alpha$.
Output: Hyperparameter vector $x^*$.

---

1: for $t = 1, 2, \ldots, T$ do.
2: Maximize the acquisition function to obtain the next evaluation point:
$x_t = \mathrm{argmax}_{x \in X}\alpha(x|D_{1:t-1})$
3: Evaluate the objective function value $y_t = f(x_t) + \varepsilon_t$;
4: Consolidate data: $D_t = D_{t-1} \cup \{x_t, y_t\}$, and update the probabilistic agent model;
5: End.

---

## 4. Experiments

### 4.1. Experimental Platform

The platform for this experiment was the operating system ubuntu 20.04, the Cuda 11.3 graphics card is RTX 3070 * 1, and the CPU is 12 core Intel core i7, 45GB RAM. PyTorch 1.11.0 deep-learning framework, Python 3.8 programming environment, and Pycharm development tools were used.

### 4.2. Model-Parameter Setting

This study used Bayesian optimization algorithm to optimize the super parameters of the model, including batch_Size, model learning rate lr, dropout, regularization coefficient, epoch, and weight-decay settings. The optimization results are shown in Table 2.

To improve the training effect and accelerate the network model's convergence, the dataset's corroded images were normalized to the normalized size of $224 \times 224 \times 3$.

The parameters generated by Image Net pre-training of VGG16 for feature extraction were retained during training, the S.E. unit module for the deflation parameter r was set to 16 as recommended by the authors of [25], and the rest of the parameters were initialized using normally distributed random values.

The value of the $\lambda$ parameter in the multi-loss-function-fusion formula was set to 0.4.

**Table 2.** Results of hyperparameter optimization.

| Hyper-Parameter | The Optimal Value |
| --- | --- |
| batch_size | 32 |
| dropout | 0.5 |
| learning rate | 0.001 |
| regularization factor | 0.5 |
| epoch | 200 |
| weight decay | 0.0005 |

*4.3. Comparison and Analysis of Experimental Results*

4.3.1. Comparison of Model Training Results

In order to verify the impact of the improved network on the accuracy of pipeline corrosion identification, the improved VGG network model in this paper is compared with the unimproved VGG model, AlexNet model, ResNet50 model, and DenseNet121 model. The experimental results are shown in Figure 11. It can be seen that when the epoch reached 125 times, these five models all achieved high accuracy, and the improved VGG model in this paper had the highest classification accuracy for the pipeline-corrosion data set.

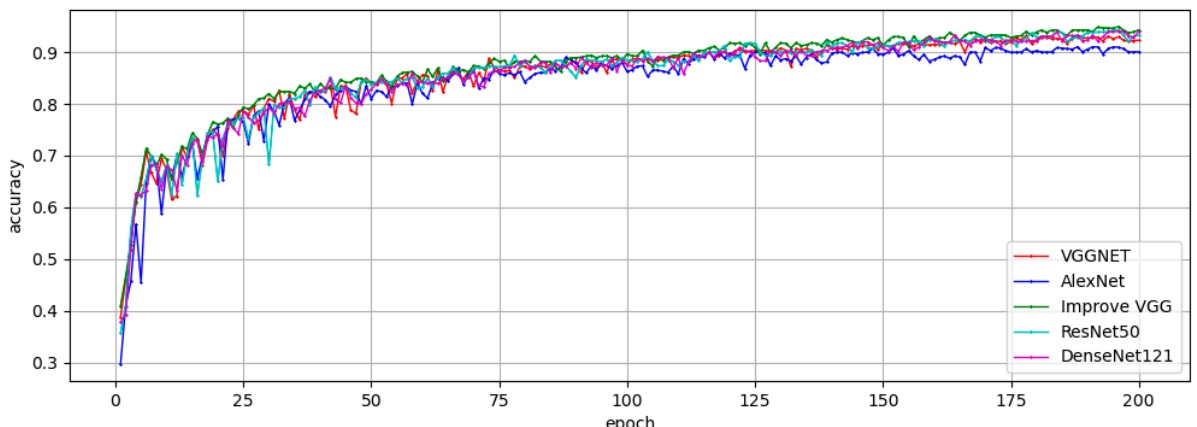

**Figure 11.** Training results of different models.

The loss-function curves of the different models are shown in Figure 12, and the loss function of the improved model relative to the others can be observed. The convolutional neural network model reached the fitted state earlier in training and gradually approached zero.

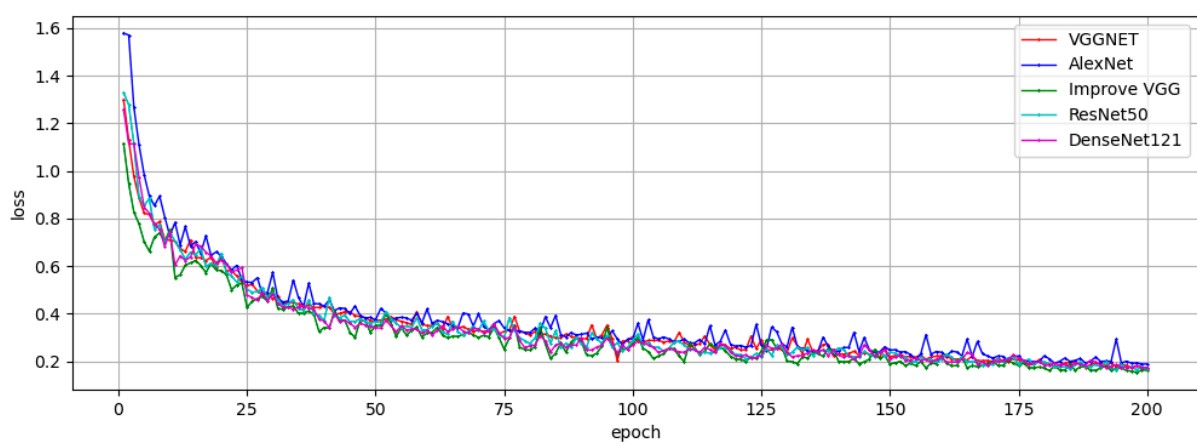

**Figure 12.** Variation of loss function for different models.

### 4.3.2. Comparison of Classification Results

The image-sample sets were divided into training and test sets and input into the improved classification model in this paper for classification and recognition. Meanwhile, to better demonstrate the recognition capability of the newly constructed classification model for water-supply pipe-corrosion detection, its performance was compared with those of the traditional VGG classifier [26], the traditional Alex model [27], the ResNet50 model [28], the DenseNet121 model [29], the SE-VGG model, and the VGG model using the joint loss function, and the confusion matrix plots of the classification and recognition results are shown in Figures 13–19.

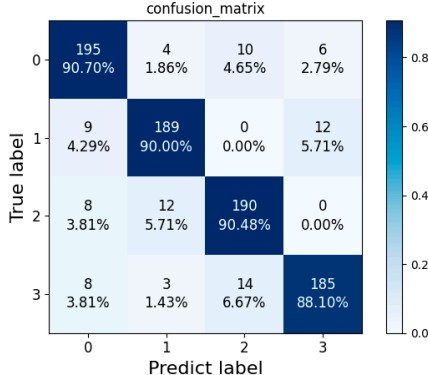

**Figure 13.** Alex classification model.

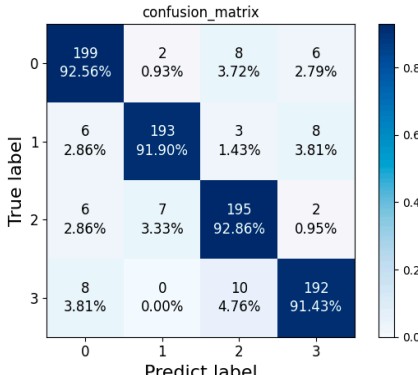

**Figure 14.** VGG classification model.

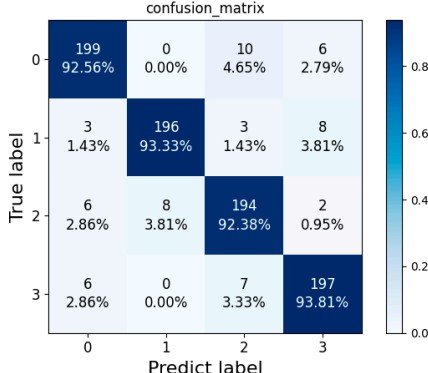

**Figure 15.** Lian-VGG classification model.

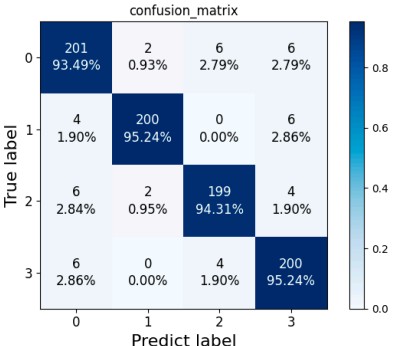

**Figure 16.** SE-VGG classification model.

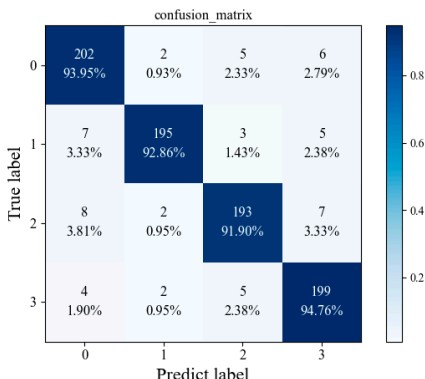

**Figure 17.** ResNet50 classification model.

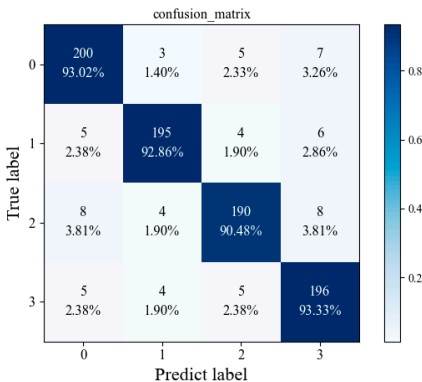

**Figure 18.** DenseNet121 classification model.

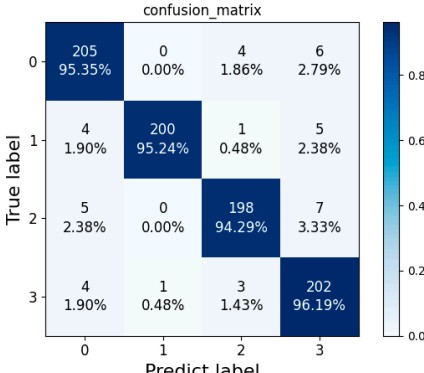

**Figure 19.** Classification model proposed in this paper.

The full name of ROC is receiver operating characteristic. The curve is drawn with the true positive rate (sensitivity) as the ordinate and the false positive rate (1-specificity) as the abscissa. For a classifier, we can get a TPR and FPR point pair according to its performance on test samples. In this way, the classifier can be mapped to a point on the ROC plane. Adjust the threshold value used by this classifier to get a curve passing (0,0), (1,1), which is the ROC curve of this classifier. The AUC value is the area below the ROC curve. Generally, AUC values range from 0.5 to 1.0, and a larger AUC represents better performance. AUC is a standard used to measure the quality of classification models. This paper uses ROC curve and AUC value to evaluate these seven models, and the results are shown in Figures 20–26.

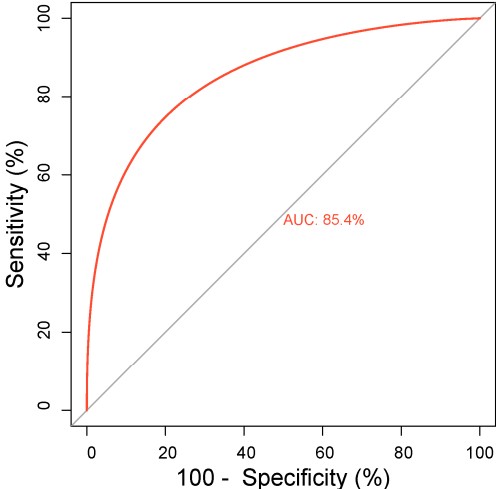

**Figure 20.** ROC Curve of AlexNet Model.

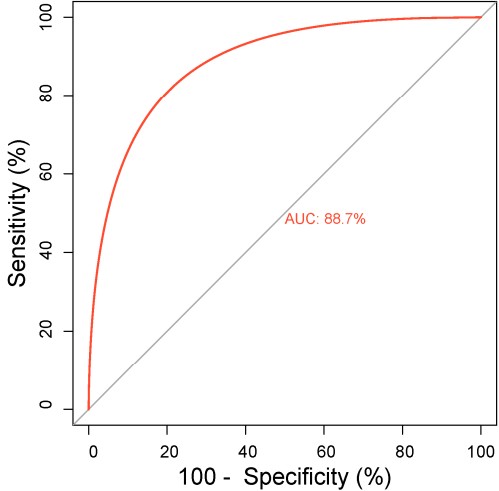

**Figure 21.** ROC Curve of VGGNet Model.

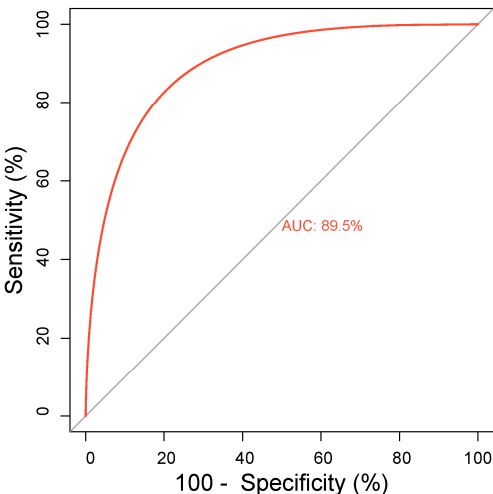

**Figure 22.** ROC Curve of Lian-VGGNet Model.

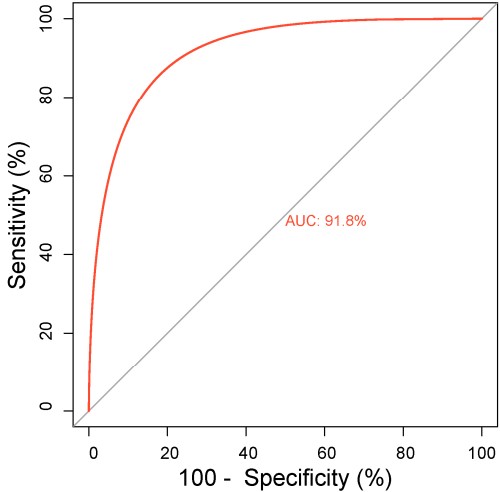

**Figure 23.** ROC Curve of DenseNet Model.

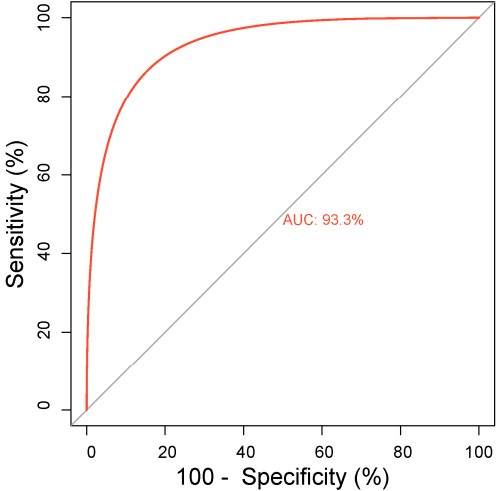

**Figure 24.** ROC Curve of ResNet Model.

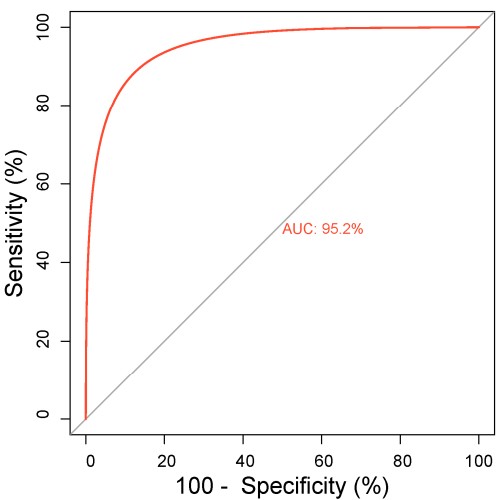

**Figure 25.** ROC Curve of SE-VGGNet Model.

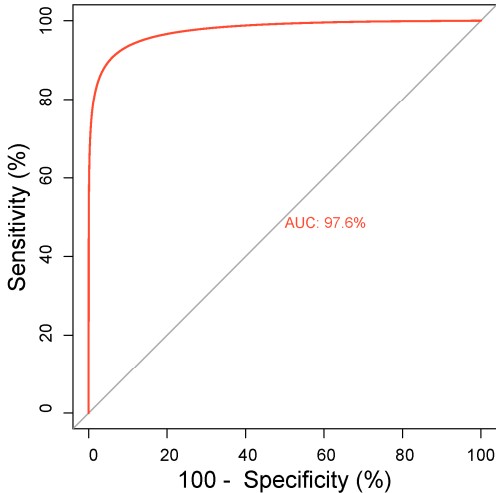

**Figure 26.** ROC curve of the improved model in this paper.

It can be seen from Figures 20–26 that the ROC curve of the improved model in this paper is closer to the optimal point (0,1). Secondly, the AUC value of the improved model in this paper is significantly higher than that of the other six comparison models, which proves the effectiveness of the model in this paper.

The above experiments achieved the identification and classification of image corrosion, following the four different types of pipe-wall-corrosion sample map under other classification models to compare and explain the classification results, as shown in Table 3. The number of normal pipe samples in the test set was 215, the number of lightly corroded pipe samples was 210, the number of pitting corrosion pipe samples was 210, and the number of surface-corrosion-pipe samples was 210.

**Table 3.** Comparison of test classification results for the four types of pipe damage sample sets.

| Corrosion Category | Alex | VGG | Lian-VGG | SE-VGG | ResNet50 | DenseNet121 | Model in This Paper |
|---|---|---|---|---|---|---|---|
| Normal pipeline | 195 | 199 | 199 | 201 | 202 | 200 | **205** |
| Slight corrosion | 189 | 193 | 196 | 200 | 195 | 195 | **200** |
| Pitting corrosion | 190 | 195 | 194 | 199 | 193 | 190 | 198 |
| Areal corrosion | 185 | 192 | 197 | 200 | 199 | 196 | **202** |
| Total number of correct | 759 | 779 | 786 | 800 | 789 | 781 | **805** |
| Classification accuracy | 89.822% | 92.189% | 93.018% | 94.674% | 93.372% | 92.426% | **95.266%** |

Next, the overall performance of the algorithm was analyzed in terms of its precision, recall, and specificity for several classification models [30], which were calculated as follows.

(1) Precision:

$$Precision = \frac{TP}{TP + FP} \tag{19}$$

(2) Recall:

$$Recall = \frac{TP}{TP + FN} \tag{20}$$

(3) Specificity:

$$Specificity = \frac{TN}{TN + FP} \tag{21}$$

where $TP, FP, TN, FN$ denote true positive, false positive, true negative, and false negative, respectively. The comparative analysis of the classification and identification results of the damage categories of the pipe's inner wall under different classifiers by the three evaluation indexes above is shown in Table 4.

**Table 4.** Comparison of test classification results for the pipe corrosion sample set.

| Corrosion Category | Classification Models | Precision | Recall | Specificity |
| --- | --- | --- | --- | --- |
| Normal pipeline | Alex | 88.636% | 90.700% | 96.031% |
| | VGG | 90.867% | 92.561% | 96.825% |
| | Lian-VGG | 92.990% | 92.561% | 97.612% |
| | SE-VGG | 92.627% | 93.492% | 97.464% |
| | ResNet50 | 91.402% | 93.953% | 96.984% |
| | DenseNet121 | 91.743% | 93.023% | 97.142% |
| | Algorithms in this paper | **94.037%** | **95.354%** | **97.936%** |
| Slight corrosion | Alex | 90.865% | 90.001% | 97.007% |
| | VGG | 95.544% | 91.903% | 98.582% |
| | Lian-VGG | 96.078% | 93.334% | 98.740% |
| | SE-VGG | 98.035% | 95.242% | 99.371% |
| | ResNet50 | 97.014% | 92.857% | 99.063% |
| | DenseNet121 | 94.660% | 92.847% | 98.307% |
| | Algorithms in this paper | **99.502%** | 95.46% | **99.842%** |
| Pitting corrosion | Alex | 88.785% | 92.381% | 96.220% |
| | VGG | 90.278% | 92.863% | 96.692% |
| | Lian-VGG | 90.654% | 92.389% | 96.850% |
| | SE-VGG | 95.215% | **94.318%** | 98.425% |
| | ResNet50 | 93.689% | 91.905% | 98.006% |
| | DenseNet121 | 93.137% | 90.476% | 97.862% |
| | Algorithms in this paper | **96.116%** | 94.295% | **98.740%** |
| Areal corrosion | Alex | 91.133% | 88.108% | 97.165% |
| | VGG | 92.307% | 91.436% | 97.480% |
| | Lian-VGG | 92.488% | 93.813% | **97.480%** |
| | SE-VGG | **92.592%** | 95.246% | 97.637% |
| | ResNet50 | 91.705% | 94.761% | 97.166% |
| | DenseNet121 | 90.323% | 93.334% | 96.692% |
| | Algorithms in this paper | 91.8188% | **96.192%** | 97.165% |

From the experimental results in Tables 3 and 4, by comparing the recognition and classification results of the Alex classification model, VGG classification model, Lian-VGG classification model, SE-VGG classification model, ResNet50 model, DenseNet121 model, and the classification model proposed in this paper, it can be seen that the classification model optimized in this paper has a better classification effect on normal, lightly corroded, and face-corroded pipes. The classification effect on point-like corrosion is less than that of the optimal algorithm. The difference between the algorithm and the optimal algorithm is

relatively small. At the same time, by comparing the accuracy, recall, and specificity of the five models, we can see that the index values of the proposed model are mostly higher than those of the other six classification models. Therefore, the improved classification model of this paper has a good effect and offers practical classification. In summary, the proposed model can provide technical support for pipeline-damage detection.

## 5. Conclusions

In this paper, based on the VGG16 network model, the S.E. attention-mechanism module was added to improve the model's ability to extract the image features in the corrosion-salience region and suppress the information related to useless features. At the same time, the loss function was improved by using the fusion of central loss function and cross-entropy loss function, which can increase the inter-class distance between different pipe-corrosion patterns and reduce the intra-class distance within the same corrosion pattern. The accuracy of pipe-corrosion classification was further improved. The final experimental results show that the proposed model achieves an accuracy of 95.266% on the pipe corrosion dataset, which is 5.444%, 3.077%, 2.248%, 0.592%, 1.894%, and 2.840% higher than the traditional AlexNet, traditional VGG, Lian-VGG, the SE-VGG models, the ResNet50 model, and the DenseNet121 models, respectively. The experimental results met the standard requirements, providing a new approach to detecting damage to the inner walls of water-supply pipes.

**Author Contributions:** Conceptualization, Q.Z. and L.L.; methodology, L.L.; software, L.Z.; validation, Q.Z., L.L. and L.Z.; investigation, L.Z.; writing—original draft preparation, L.L.; writing—review and editing, Q.Z.; visualization, L.L. and L.Z.; supervision, Q.Z. All authors have read and agreed to the published version of the manuscript.

**Funding:** This research was supported by the National Natural Science Foundation of China (Grant No. 51804248), the Shaanxi Provincial Science and Technology Department Industrial Research Project (Grant No. 2022GY-115), the Beilin District Applied Technology R&D Project (Grant No. GX2114), and the Shaanxi Provincial Education Department Service to Local Enterprises (No. 22JC050).

**Institutional Review Board Statement:** Not applicable.

**Informed Consent Statement:** Not applicable.

**Data Availability Statement:** Data is available on request from the corresponding author.

**Conflicts of Interest:** The authors declare no conflict of interest.

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
