# Peer review of "Identification of Corrosion on the Inner Walls of Water Pipes Using a VGG Model Incorporating Attentional Mechanisms"

_applsci, doi:10.3390/app122412731_

Round 1

Reviewer 1 Report

This paper proposes an improved VGG model for classifying corrosion defect on the inner wall of water pipes. It is an interesting topic to address some practical problems in water pipe defect detection. However, there are still some shortcomings in this paper.

(1) The authors improved the VGG model only using the S.E attention mechanism. However, the attention mechanism is very popular. The authors have not improved it in any way. Therefore, the article is not innovative enough.

(2) In the experiment, the authors only compared the improved method with the AlexNet model. More models such as ResNet and DenseNet should be compared.

(3) In section 2, the section title is Related work. However, the content is not related work. It's more like the generation of data sets.

(4) On page 5, there are two formulas numbered 1. The formula in line 154 is wrong.

(5) There are many typos and sentence/grammatical errors in the work, which are unforgivable at this level of scholarly work. For example, line 237 page 8, ...and the improved Improve VGG model.....; line 195 page 7, x is the input,

(6) For the format of the paper, all the margins of each line should be aligned.

Reviewer 2 Report

This paper presented a convolutional neural network-based identification method for detecting the corrosion of inner walls of water pipes under VGG models. The proposed approach was a vision-based method using image processing capability of NNs. The research topic is interesting and within the main scope of the journal. However, the following major comments should be addressed in the manuscript. 

 1. The introduction section should contain a flow at which the reason of implementing the research is clearly explained. 

2. The mathematical concept of the CNNs should be explained with more details. 

3. The main idea behind using ML algorithms should also be discussed (see  structural control and health monitoring vol. 28 issue 11, e2825). 

4. It is more appropriate if an optimization algorithm be used to tune the hyperparameters of the ML tools. If it is not possible the authors should at least address this issue (see  "A rapid neural network-based demand estimation for generic buildings considering the effect of soft/weak story). 

5. The results should contain the ROC curves for each confusion matrix presented in the paper. 

Round 2

Reviewer 1 Report

The formula in line 291 is still wrong. In this formula, x is not the input value, it is the index symbol for the sum and it should appear at the subscripts of some symbols in the formula.

Reviewer 2 Report

The authors have addressed all the comments provided by the authors.  The paper is ready to be published. 
